# Characterization of nanomaterials synthesized from *Spirulina platensis* extract and their potential antifungal activity

Agnieszka Sidorowicz[1,2]☉, Valentina Margarita[3]☉, Giacomo Fais[1,2], Antonella Pantaleo[3], Alessia Manca[3], Alessandro Concas[1,2], Paola Rappelli[3,4], Pier Luigi Fiori[3,4]*, Giacomo Cao[1,2]*

**1** Interdepartmental Centre of Environmental Science and Engineering (CINSA), University of Cagliari, Cagliari, Italy, **2** Department of Mechanical, Chemical and Materials Engineering, University of Cagliari, Piazza d'Armi, Cagliari, Italy, **3** Department of Biomedical Sciences, University of Sassari, Sassari, Italy, **4** Mediterranean Center for Disease Control, Sassari, Italy

☉ These authors contributed equally to this work.
* fioripl@uniss.it (PLF); giacomo.cao@unica.it (GC)

**Data Availability Statement:** All relevant data are within the article and its Supporting Information files.

## Abstract

Nowadays, fungal infections increase, and the demand of novel antifungal agents is constantly rising. In the present study, silver, titanium dioxide, cobalt (II) hydroxide and cobalt (II,III) oxide nanomaterials have been synthesized from *Spirulina platensis* extract. The synthesis mechanism has been studied using GCMS and FTIR thus confirming the involvement of secondary metabolites, mainly amines. The obtained products have been analysed using XRD, SEM, TGA and zeta potential techniques. The findings revealed average crystallite size of 15.22 nm with 9.72 nm for oval-shaped silver nanoparticles increasing to 26.01 nm and 24.86 nm after calcination and 4.81 nm for spherical-shaped titanium dioxide nanoparticles which decreased to 4.62 nm after calcination. Nanoflake shape has been observed for cobalt hydroxide nanomaterials and for cobalt (II, III) oxide with crystallite size of 3.52 nm and 13.28 nm, respectively. Silver nanoparticles showed the best thermal and water dispersion stability of all the prepared structures. Once subjected to three different *Candida* species (*C. albicans*, *C. glabrata*, and *C. krusei*) silver nanoparticles and cobalt (II) hydroxide nanomaterials showed strong antifungal activity at 50 μg/mL with minimum inhibitory concentration (MIC) values. After light exposition, MIC values for nanomaterials decreased (to 12.5 μg/mL) for *C. krusei* and increased (100 μg/mL) for *C. albicans* and *C. glabrata*.

## Introduction

In the last two decades, the number of fungal infections significantly increased due to a higher prevalence of immunocompromised patients, broad-spectrum use of antibiotics, and growing need for hyperalimentation in hospitals [1]. One of the most common fungal infections is candidemia, caused by *Candida* spp, responsible for 15% of nosocomial infections, with an estimated mortality of 47% [2]. In addition, the risk of infection increases when surgical and

**Funding:** The authors received no specific funding for this work.

**Competing interests:** The authors have declared that no competing interests exist

mechanical devices are used, such as endotracheal tubes, drains, or urinary catheters [3]. Other potential risk factors include lack of, or inadequate, hygiene-environment of healthcare workers and duration of stay in the intensive care unit for more than 7 days [4, 5].

Currently, treatment of fungal infections is based on polyenes, azoles, or echinocandins, which demonstrate strong fungicidal activity but can also cause adverse side effects or trigger several additional medications [6]. Moreover, such products could be poorly tolerated or could show a narrow spectrum of activity [7]. Recently, polyenes, azoles, and echinocandins have displayed an increase in *Candida* resistance due to their general and long-term use [8, 9].

As for new fungicidal agents significant attention has been given to nanomaterials (NMs) owing to their wide range of applications. Synthesized nanostructures have a size range of 1–100 nm in at least one dimension and altered physicochemical properties compared with bulk material including increased surface to volume ratio, high reactivity, altered magnetic and optical properties. Commonly obtained antimicrobial nanoparticles (NPs) include Ag NPs [10], ZnO NPs [11], CuO NPs [12], $TiO_2$ NPs [13], and Au NPs [14]. However, albeit showing strong antimicrobial activity, most of the synthesized NMs cannot be employed due to their low biocompatibility, especially when synthesized using chemical methods [15–17]. In this regard, green synthesis routes have been shown to be effective, simple, and affordable to produce biocompatible NMs [18]. Specifically, the techniques based on the use of extracts from blue-green algae seem quite promising [19].

It should be noticed that several works have investigated the synthesis of silver nanoparticles (Ag NPs), including the so called "allotropic silver" form [20], and their applications. In particular, Ismail and colleagues studied antioxidant, antiviral and antimicrobial activity of Ag NPs synthesized using phycobiliprotein extract of *S. platensis* [21]. The results showed the great potential of *S. platensis* extract as a tool to synthesize biocompatible Ag NPs to be used in the biomedical field.

Along with Ag NP, titanium dioxide NPs ($TiO_2$ NPs) are materials of great significance, mainly used for the photodegradation of several environmental contaminants [22, 23]. Moreover, low toxicity and good biocompatibility of $TiO_2$ NPs find their applications in the biomedical field, such as bone tissue engineering, targeted drug delivery systems, or antibacterial devices for prevention and treatment of infections [24–26].

Recently, cobalt hydroxide ($Co(OH)_2$ NMs) and cobalt oxide nanomaterials ($Co_3O_4$ NMs) have been tested for biological as well as magnetic and catalytic applications [27]. Currently, to the best of our knowledge no reports are available on the synthesis of cobalt nanomaterials using *Spirulina* extract. Although Anwar et al describe the use of $Co(OH)_2$ NPs therapy against *Acanthamoeba castellani* infections, antimicrobial properties of green-synthesized $Co(OH)_2$ NMs have not been studied so far [28]. It is also worth mentioning that the photoactivity of NMs for a degradation of organic pollutants has been also investigated [29].

In our study, *S. platensis* extract has been used to synthesize six types of NMs: Ag NPs (before and after calcination), $TiO_2$ NPs (before and after calcination), $Co(OH)_2$ NMs, and $Co_3O_4$ NMs, in different concentrations with the presence of absence of light for their antifungal activity against *Candida albicans*, *C. glabrata*, and *C. krusei*. The participation of secondary metabolites from the extract was assessed and the characterization of the NMs was carried out with the aim of fully understanding the nanomaterials properties along with their antifungal activity.

## Materials and methods

### Preparation of the extract

Unialgal culture of cyanobacterium *S. platensis* was obtained from the operating plant of TOLO Green located in Arborea (Sardinia, Italy) and cultivated in modified Zarrouk's

**Table 1. Chemical composition of Zarrouk Medium and modified Zarrouk Medium.**

| Components in modified Zarrouk medium | Concentration (g/L) | Components in Zarrouk medium | Concentration (g/L) |
|---|---|---|---|
| $NaHCO_3$ | 7.0 | $NaHCO_3$ | 16.8 |
| $NaNO_3$ | 3.0 | $NaNO_3$ | 2.5 |
| $CaCl_2 \cdot 2\,H_2O$ | 0.04 | $CaCl_2$ | 0.04 |
| $K_2HPO_4$ | 0.8 | $K_2HPO_4$ | 0.5 |
| NaCl | 1.0 | NaCl | 1.0 |
| $Na\text{-}EDTA\cdot 2\,H_2O$ | 0.1 | $Na\text{-}EDTA\cdot 2\,H_2O$ | 0.08 |
| $FeSO_4 \cdot 7\,H_2O$ | 0.01 | $FeSO_4 \cdot 7\,H_2O$ | 0.01 |
| $K_2SO_4$ | 0.4 | $K_2SO_4$ | 1.0 |
| $MgSO_4$ | 0.24 | $MgSO_4$ | 0.2 |
| KOH | 1.0 | | |

Medium for enhanced biomass production [30]. The medium was prepared according to the composition in Table 1 while $K_2SO_4$ and $MgSO_4$ were added aseptically after setting pH to 9 and autoclaving the resulting solution under high pressure at 121˚C for 20 min. It should be noted that the trace metal solution Zarrouk medium had the following composition: $H_3BO_3$ 2860 mg/L, $ZnSO_4\cdot 7\,H_2O$ 222 mg/L, $MnCl_2\cdot 4\,H_2O$ 1810 mg/L, $CuSO_4\cdot 5\,H_2O$ 79 mg/L while the trace metal solution Modified Zarrouk Medium was characterized by the following composition: EDTA 250 mg/L, $H_3BO_3$ 57 mg/L, $ZnSO_4\cdot 7\,H_2O$ 110 mg/L, $MnCl_2\cdot 4\,H_2O$ 25.3 mg/L, $CoCl_2\cdot 6\,H_2O$ 8.05 mg/L, $CuSO_4\cdot 5\,H_2O$ 7.85 mg/L, $Mo_7O_{24}(NH_4)_6\cdot 4\,H_2O$ 5.5 mg/L.

The culture was cultivated in aerated photobioreactor (Medinlab™, Italy) under continuous illumination of white and blue LED lamps (Nicrew®) that ensured a photosynthetic photon flux density (PPFD) of 30 μmol m$^{-2}$ s$^{-1}$. A picture of the entire process to obtain nanomaterials from *S. platensis* extract is reported in Fig 1. The temperature of the environment was set to 25˚C. The initial optical density at a wavelength of 750 nm ($OD_{750}$) was equal to 0.05 and increased to 0.4 after 14 days of cultivation. Then, the culture was centrifuged at 4˚C under 1500 RPM (Heraeus® Megafuge® 1.0R) and the residue was dried at room temperature for 7 days. Next, the dried residue was weighted, and 5 g of biomass were added to 300 mL of methanol (Merck® LiChrosolv® hypergrade). The flask was sonicated for 30 min (Soltec® Sonica® 2400 ETH S3) and stirred at 250 RPM (IKA® RH Digital Magnetic Stirrer) for additional 30 min. To remove the biomass, the suspension was filtered using standard filtration paper (Whatman®) and then evaporated using rotary evaporator (BUCHI Rotavapor™ R-210 Rotary Evaporator System) to remove about 70% of methanol. The concentrated extract was then diluted to the final volume of 600 mL and divided in three equal aliquots to be used for the synthesis.

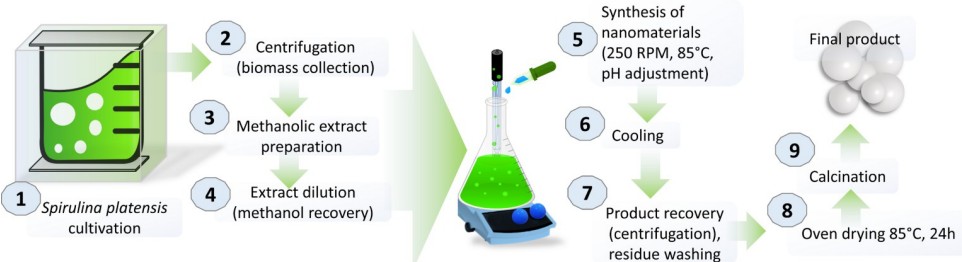

**Fig 1. Preparation of nanomaterials from *S. platensis*.**

## Synthesis of nanostructures

The prepared extract was heated to 85˚C and stirred at 250 RPM (IKA® RH Digital Magnetic Stirrer) and then a 2 ml sample was taken and labelled as "before the synthesis". Subsequently, metal salts in powder were added to the extract to obtain 0.1 M solution of silver nitrate (Carlo Erba®), cobalt (II) chloride hexahydrate (Carlo Erba®) and titanium (IV) oxysulfate—sulfuric acid hydrate (AlfaAesar®) (3.40 g, 4.76 g and 5.56 g respectably). Next, after 15 min, the pH was raised by 3–4 points (for silver and cobalt salts up to 8, for titanium up to 4) using 1.25 M NaOH. After the salt has been added, the reaction (with stirring and heating) takes place for 1.5 h since it is quenched by removing from heat source for about 30 min. Subsequently, the resulting solution was centrifuged at 8˚C using 4000 RPM (Heraeus® Megafuge® 1.0R) and the obtained supernatant was withdrawn and labelled as "after the synthesis", while the residue was repeatedly washed with MiliQ $H_2O$ (Millipore®, Milan, Italy) in subsequent centrifugation cycles. Finally, the residue was dried at 80˚C for 24 h and then ground using mortar and pestle to obtain two equal parts. One part was stored in Eppendorf tube in absence of light and the other one was calcined in muffle furnace (Gelman Instrument®) for 3 h at different temperatures (150˚C for silver, 400˚C for titanium and 450˚C for cobalt). After calcination, the samples were stored under the same conditions as the samples before calcination.

## Materials characterization

To investigate the influence of operating conditions such as pH and elevated temperature, an additional experiment was performed (mimicking reaction). The extract was heated to 85˚C and stirred at 250 RPM (IKA® RH Digital Magnetic Stirrer). Then, a 1.25 M HCl solution was added to obtain the same pH as the flasks used during the synthesis process previously mentioned (5 for silver and cobalt, 0 for titanium). Subsequently, after 15 min pH was raised using a solution 1.25 M NaOH to the same values reached during the synthesis process (8 for silver and cobalt, 4 for titanium). Since the moment of adding diluted acid, the reaction continued for 1.5 h and then was left to cool down (the exact protocol as synthesis of nanomaterials). At the end point, the samples from mimicking reactions were taken from both flasks and labelled for GCMS sample preparation.

During sample preparation, 400 μl of the liquid samples taken before the synthesis, after mimicking reaction occurs and the synthesis of Ag NPs as well as $TiO_2$ NPs were dried under nitrogen flow. Then, they were derivatized with 100 μL of methoxyamine hydrochloride/pyridine solution (Sigma Aldrich, Milano, Italy) (Sigma Aldrich, Milano, Italy). The samples were vortexed and incubated for 17 h. Subsequently, 100 μL of NO-bis (trimethylsilyl) trifluoroacetamide (BSTFA) (Sigma Aldrich, Milano, Italy) was added and samples were vortexed. Next, 800 μL of 20 ppm 2-dodecanone/hexane (Sigma Aldrich, Milano, Italy) solution was mixed with the liquids. At the end of the preparation step, the samples were filtered with 0.45 μm syringe filter (Sigma Aldrich, Milano, Italy). The sample after cobalt synthesis (prepared separately to avoid precipitation), once transferred to glass vial (400 μl), was dried under nitrogen flow, then 200 μL of BSTFA was added. The sample was finally vortexed, mixed with 800 μL of 20 ppm 2-dodecanone/hexane and filtered to avoid precipitation.

After derivatization, samples were injected in a Hewlett Packard 6850 Gas Chromatograph, 5973 mass selective detector (Agilent Technologies, Palo Alto, CA), using helium as carrier gas at 1.0 mL/min flow. 1 μL of each sample was injected in the split-less mode and resolved on a 30 m × 0.25 mm × 0.25 μm DB-5MS column (Agilent Technologies, Palo Alto, CA). Inlet, interface, and ion source temperatures were 250, 250 and 230˚C, respectively. The starting temperature of the oven was set to 50˚C while the final one was led to 230˚C using a heating rate of 5˚C/min for 36 min and then the sample was kept at a constant temperature for 2 min.

Electron impact mass spectra were recorded from 50 to 550 m/z at 70 eV. The identification of metabolites was performed by mass spectra comparison with analytical standards using the NIST14 library database of the National Institute of Standards and Technology (Gaithersburg, MD).

The Fourier transform infrared spectroscopy (FTIR) was performed using Nicolet™ iS™ 10 FTIR Spectrometer (Thermo Fisher Scientific®, Madison, Wisconsin, USA) in the range 400–4000 $cm^{-1}$.

X-Ray Diffraction (XRD) analysis of the prepared nanostructures was carried out using X-Ray Diffractometer (Phillips®, PW1830/00, Nederlands and Rigaku SmartLab® with graphite monochromator in the diffracted beam). The scanning was performed at 40 kV and 30 mA using CuKα (λ = 1.54 Å) radiation with a diffraction angle between 12˚ and 90˚ using 0.1˚ interval within 10 s. Diffraction pattern from each sample was analysed with HighScore Plus (v 3.0) and Maud™ (v. 2.99) software as well as Crystallography Open Database (COD, October 2014). Crystallite size was estimated after baseline correction using the well-known Debye-Scherrer's formula.

Scanning Electron Microscopy (SEM) analysis was carried out using Hitachi S4000 FEG HRSEM (Hitachi Ltd., Tokyo, Japan) operated at 20 kV. Image acquisitions were obtained by Quartz PCI software (Quartz Imaging Corporation, Vancouver, Canada). The samples were coated with 2 nm of platinum to enhance the contrast. EDX analysis results were acquired using UltraDry EDX Detector (Thermo Fisher Scientific®, Madison, Wisconsin, USA), and NSS3 software (Thermo Fisher Scientific®, Madison, Wisconsin, USA). The image analysis was performed by ImageJ 1.52a software.

Thermogravimetric Analysis (TGA) was performed using Differential Thermal Analyzer TG/DSC (NETZSCH® STA 409 PC) in a temperature range 25–1000˚C and a heating rate of 10˚C/min with air flow 100 mL/min.

Zeta potential measurement was performed using Zetasizer® Helix (Malvern Panalytical Ltd., UK). Before the measurement, the samples were sonicated using a suitable bath (Branson Ultrasonics™ 1210, USA) for 20 min. The results were analysed using Zetasizer® Software (Malvern Panalytical Ltd., UK).

## Antifungal study

Three Candida species were used in this work: *C. albicans*, *C. glabrata* and *C. krusei*. All microbial strains were previously isolated from vaginal swabs of patients with fungal infection and collected from the laboratory of Microbiology of University of Sassari (Italy).

Microorganisms were isolated in selective chromogenic culture solid media and then characterized using standard mass spectrometry identification techniques (MALDI-TOF). The strains were stored at -80˚C until use; microbial strains were thawed in Luria Bertani (LB) Medium (Merck) and were grown overnight at 37˚C under shaking (180 RPM). After 24 h, the growth of each isolate was assessed spectrophotometrically at 600 nm.

Antifungal activity of nanomaterials was assessed by microdilution method, using 48 wells plates. Nanomaterials were suspended in 1 ml of Phosphate Buffer Saline (PBS) 1X (Merck) sterile, vortexed and sonicated 1 min three times to remove debris that could interfere with assay. Serial two-fold dilutions of each nanomaterial, ranging from 100 μg/ml to 6 μg/ml, were performed in PBS 1x in the rows of a 48 wells-plates.

*C. albicans*, *C. glabrata* and *C. krusei* were seeded at $5 \times 10^4$ cell/well and incubated at 37˚C and 5% $CO_2$ in the various concentrations of nanomaterials and extract before the synthesis for 48 h in the presence or absence of light. *Candida spp*. in LB broth added with amphotericin B (16 μg/ml) [31] was used as positive control while strains in LB broth were used as negative control. LB broth was used as blank.

After 48 hours of incubation, the plates were microscopically observed and the minimum inhibitory concentration (MIC), defined as the lowest drug concentration that inhibit the visible growth of a microorganism at the end of the incubation period, was detected [32]. To determinate whether the nanomaterial concentration could inhibit the growth of fungi, the entire volumes of wells with the highest inoculum were seeded in LB agar plates (100 μL aliquots/plate) which were incubated at 37°C for 48 hours. The nanomaterial was evaluated to be fungicidal when it was able to kill ≥99.9% of the final inoculum. Moreover, the effect of light on viability of microorganisms was evaluated by studying the growth of *Candida* species under light exposition.

The experiments were carried out in triplicate. The significance was calculated using Student's *t*-test and difference were considered significant at $p < 0.05$.

## Results and discussion

*S. platensis* is a well-known source of various nutrients which are used as food supplements worldwide. Moreover, extracted secondary metabolites can play an important role not only in human nutrition but also for the synthesis of nanomaterials. The exact composition of the algal extract (before synthesis, mimicking reaction in pH 4, mimicking reaction in pH 8, after synthesis of Ag NPs, after synthesis of $TiO_2$ NPs and after synthesis of $Co(OH)_2$ NMs), as described in the methodology, can be found in S1–S3 Tables.

By comparing the composition of extracts before synthesis, mimicking and after synthesis it can be inferred that in all cases amines such as methylamine, diethanolamine, and ethanamine are clearly involved in the synthesis of NPs. The observed results agree with the existing literature. Amines have been used in the past for nanoparticle synthesis using chemical synthesis route [33]. Recently, they are also used to synthesize and functionalize nanoclusters and could play a role in obtaining their ultrasmall size [34, 35]. Diethanolamine was used for the synthesis of silver and titanium-based nanoparticles working as a reaction controlling agent to form homogeneous precipitates and in case of silver decreased its size more efficiently than mono-ethanolamine used during the reaction [36, 37]. The study highlighted the important role of hydroxyl groups present in diethanolamine which can act as reducing agents. In additions, when studying the synthesis mechanism using amines of cobalt oxide nanomaterials, it was observed that initially three-dimensional (3D) nanoparticles are formed from the precursor solution and then they start transforming into 2D nanosheets at critical size of 3.0–4.4 nm [38]. Moreover, the authors proved the interactions of cobalt oxide with $-NH_2$ group using methylamine model. Most probably amines during the synthesis acted as reducing agents (Fig 2) that provide electrons to metal ions facilitating the formation of metal NPs as well as the attachment of other secondary metabolites which act as stabilizing (capping) agents that

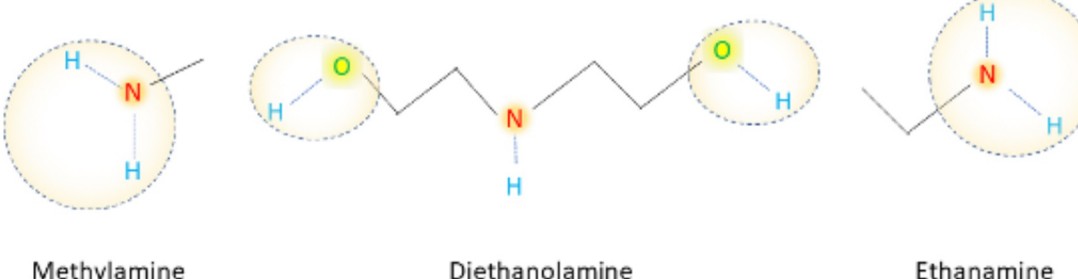

Methylamine                    Diethanolamine                    Ethanamine

**Fig 2. Main functional groups in methylamine, diethanolamine, and ethanamine, respectively participating in the reduction of metal ions into nanomaterials.**

hinder the surface reaction of newly formed NPs with compounds in the surrounding environment.

During the Ag NPs synthesis in addition to amines, ethyl mandelate, propylene glycol, butanedioic acid, L-glutamic acid derivative, and glyceryl-glycoside derivative were utilized. Although the role of ethyl mandelate was not studied before, it may act as a capping agent. The properties of propylene glycol were studied mainly as an emollient in Ag NPs loaded hydrogels to increase their stability [39]. Butanedioic acid on the surface of Ag NPs was tested for sensitive quantification of melamine [40] and, in the present study it was probably attached to the amine ligands. Although the involvement of L-glutamic acid derivatives in the Ag NPs synthesis was not empirically verified, due to its antioxidant properties, it can be reasonably assumed that it plays a key role in the further reduction of silver metal ions into silver NPs. Similar role can be attributed to glyceryl-glycoside derivative, which was reported to undergo hydrolysis and oxidation during the formation of Ag NPs [41]. Interestingly, during the reaction, concentration of the glyceryl-glycoside derivative isomer is increasing as well as of 2-methylpentan-2-ol while the exact mechanism of their action is unknown.

The important role of proteins present in the *S. platensis* extract for the synthesis of Ag NPs was investigated by Ameen et al [42]. In addition to polysaccharides from *S. platensis* extract, Attia et al utilized gamma radiation to synthesize Ag NPs [43]. The product was successfully used against *Erwinia amylovora*, Gram-negative bacteria, which caused fire blight infection on trees. Results showed that both proteins and polysaccharides from *S. platensis* on the surface of Ag NPs have prominent antimicrobial effects.

Compared with mimicking reaction, in the TiO$_2$ NPs synthesis, lactic acid and lactose were utilized apart from amines. Beyond acting act as an acid catalyst and electron donor, lactic acid was reported to mediate surface interactions of nuclei favouring the coalescence of nanoparticles formed at the beginning of the synthesis, during hydrolysis and condensation [44]. The study described binding of the Ti(IV) ions by a donor-acceptor bond, which transfers the electrons and reduces its partial positive charge. Therefore, Ti(IV) ions' reactivity and equilibrium constant (during heating) of hydrolysis and condensation are reduced with lactic acid acting as a capping ligand further mediating formation of nanoparticles. While the involvement of lactose during TiO$_2$ NPs was not described so far, owning to its reducing properties, it may participate as a reducing agent. Similarly, to the Ag NPs synthesis, there is also a contribution of glyceryl-glycoside derivatives and 2-methylpentan-2-ol in addition to increasing concentration of glycerol.

So far, *S. platensis* was used to synthesize TiO$_2$ NPs while TiO$_2$ was added directly to the culture, and the product was retrieved after one week [45]. The authors observed nanoparticles' penetration into the *Spirulina* cells, thus resulting in a notch-like structure with consequent cell damage. The antimicrobial application of TiO$_2$ NPs was foreseen, however not tested, and the effects of *S. platensis* metabolites on TiO$_2$ NPs formation remained unknown.

Additional secondary metabolites were observed to be utilized during Co(OH)$_2$ NMs synthesis such as amines, ethyl mandelate, 1,4-bis(trimethylsilyl) benzene, nonane, ethylene glycol, propylene glycol, lactic acid, glycerol, butanedioic acid, L-glutamic acid derivative, glyceryl-glycoside derivatives, and lactose. The role of ethyl mandelate, propylene glycol, lactic acid, butanedioic acid, L-glutamic acid derivative, glyceryl-glycoside derivative and lactose is probably similar to their action during synthesis of Ag NPs and TiO$_2$ NPs. Ethylene glycol and glycerol may act as stabilizing agents comparably with propylene glycol. The exact role of 1,4-bis(trimethylsilyl)benzene and nonane is unknown. Interestingly, the sample after synthesis also consisted of propylamine and retinal, however, description of their involvement requires further studies.

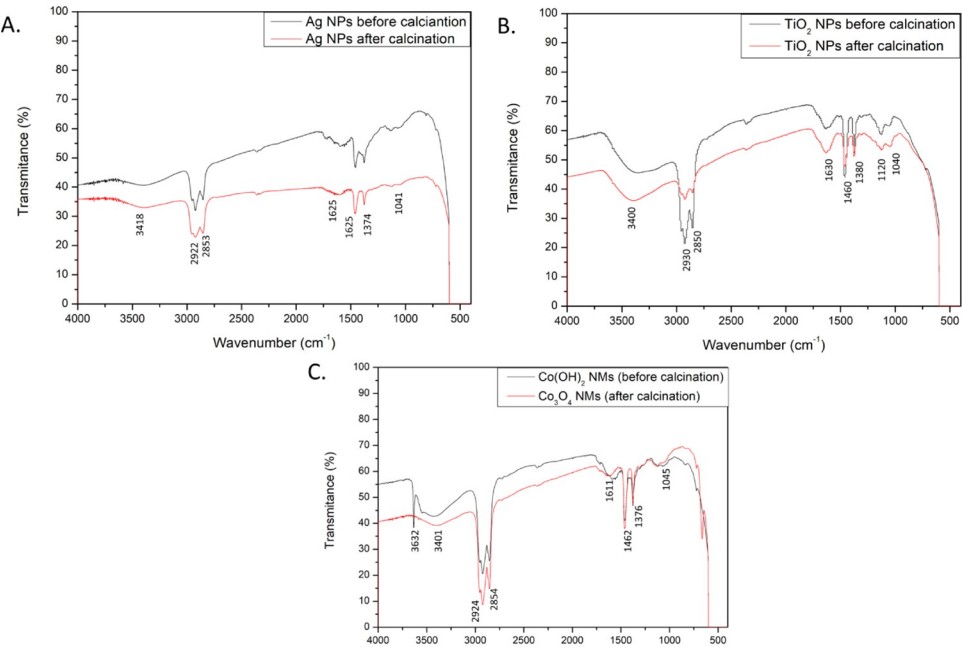

**Fig 3. FTIR spectra of the prepared nanomaterials.** (A) Ag NPs before and after calcination, (B) TIO2 NPs before and after calcination, (C) Co(OH)2 NMs and Co3O4 NMs.

The synthesis of cobalt-based nanocatalysts has been investigated so far using levulinic acid, gallic acid, starch, and metal-organic substrates to evaluate their catalytic activity towards dyes [46]. In addition, $Co_3O_4$ NMs prepared by using *Geranium wallichianum* and *Populus ciliate* plants leaves extract showed antimicrobial properties [47, 48]. In such investigation, the involvement of stigmasterol, ursolic acid, β-sitosterol, β-sitosterol galactoside, and herniarin in the $Co_3O_4$ NMs synthesis was assumed albeit no experiments have been conducted to elucidate the corresponding mechanism.

The involvement of capping agents on the surface of nanomaterials was further confirmed using FTIR technique (Fig 3). The results revealed the presence of hydroxyl groups in high concentration on the surface of Ag NPs at 3418 cm$^{-1}$ (Fig 3A). Moreover, alkyl -CH$_2$- groups were found at 2922 cm$^{-1}$ and 2853 cm$^{-1}$ with P-C organophosphorus compounds at 1463 cm$^{-1}$. Attachment of amines was confirmed at 1625 cm$^{-1}$ (NH group, probably belonging to diethanolamine) and at 1041 cm$^{-1}$ (C-N group of aliphatic amines) which also supports GCMS findings. Reducing action of amines was corroborated by the presence of NO group at 1374 cm$^{-1}$. Abundance of halides and carbon-metal bonds was confirmed at 550–400 cm$^{-1}$. Analogous FTIR pattern was obtained for other Ag NPs synthesized from *S. platensis* [49].

Similarly, for TiO$_2$ NPs hydroxy groups were detected at 3400 cm$^{-1}$ with alkyl -CH$_2$- groups at 2930 cm$^{-1}$ and 2850 cm$^{-1}$ (Fig 3B). Organophosphorus P-C group, belonging possibly to phospholipids or proteins, was confirmed at 1460 cm$^{-1}$ and C-O groups were identified at 1120 cm$^{-1}$. Functional groups of amines were detected at 1630 cm$^{-1}$ and 1040 cm$^{-1}$ belonging to NH and CN groups respectably. Nitro compounds with NO groups were noted at 1380 cm$^{-1}$. Low transmittance signal for 550–400 cm$^{-1}$ suggests the presence of halides and carbon-metal bonds in high concentration.

The functional groups present on the surface of Co(OH)$_2$ NMs and Co$_3$O$_4$ NMs were comparable with the ones found in the nanomaterials discussed so far (Fig 3C). In particular, hydroxyl groups in abundance were detected at 3401 cm$^{-1}$ with carbonyl groups at 2924 cm$^{-1}$ and 2854 cm$^{-1}$.

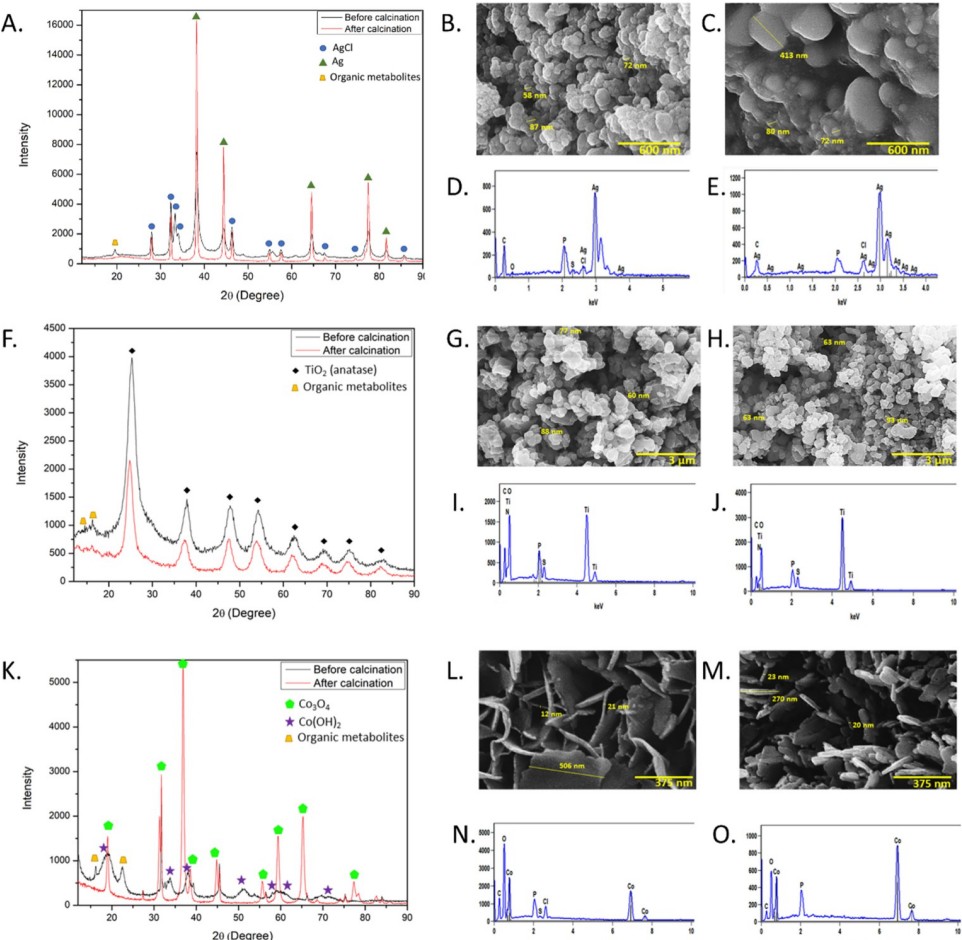

**Fig 4. Crystallographic and microscopic studies of prepared nanomaterials.** (A) XRD spectrum of Ag NPs, (B) SEM image of Ag NPs before calcination, (C) SEM image of Ag NPs after calcination, (D) EDX of Ag NPs before calcination, (E) EDX of Ag NPs after calcination, (F) XRD spectrum of TiO2 NPs, (G) SEM image of TiO2 NPs before calcination, (H) SEM image of TiO2 NPs after calcination, (I) EDX of TiO2 NPs before calcination, (J) EDX of TiO2 NPs after calcination, (K) XRD spectrum of Co(OH)2 NM and Co3O4 NM, (L) SEM image of Co(OH)2 NM (before calcination), (M) SEM image of Co3O4 NM (after calcination), (N) EDX of Co(OH)2 NM (before calcination), (O) EDX of Co3O4 NM (after calcination).

Organophosphorus P-C groups were found at 1462 cm$^{-1}$. Attachment of amines was confirmed at 1611 cm$^{-1}$ and 1045 cm$^{-1}$ belonging to NH and CN groups, respectively. Nitro compounds were identified at 1377 cm$^{-1}$. Only in the case of Co(OH)$_2$ NMs, the peak at 3632 cm$^{-1}$ was caused by stretching vibration of OH group in its structure. Like the other synthesized nanomaterials, there is a visible abundance of halides and metal-carbon bonds at 550–400 cm$^{-1}$.

FTIR results show similar pattern of functional groups acting as capping agents during the reaction. The findings further confirm involvement of secondary metabolites, especially amines, as active compounds for the synthesis of various nanomaterials.

The XRD patterns are presented in Fig 4. Both Ag NPs before and after calcination show AgCl and Ag phases in their structure with possibly attached organic metabolites (Fig 4A). The product after calcination shows increased intensity of peaks corresponding to Ag phase. The findings were matched with three phases in Crystallography Open Database (COD) with database ID codes: 9011666, 9011673, and 1100136 corresponding to cubic AgCl, monoclinic AgCl, and cubic Ag, respectively. The average crystallite size estimated using the Debye-

Scherrer's equation was 15.22 nm for AgCl phase and 9.72 nm for Ag before calcination with increase after calcination to 26.01 nm for AgCl and 24.86 nm for Ag.

It should be noted that presence of both Ag and AgCl phases was confirmed by using *Scenedesmus sp* green microalgae culture for intracellular production of NPs while their crystallite size was not evaluated [50]. Current research on Ag NPs utilizing *S. platensis* confirmed the presence of Ag phase [20, 21]. The additional occurrence of AgCl in the present study indicates the key role of secondary metabolites from extract in the reaction mechanism leading to the synthesis of NPs.

In case of $TiO_2$ NPs (Fig 4F), the observed XRD pattern matches tetragonal $TiO_2$ anatase (COD ID code 9008215) with probably secondary metabolites attached. After calcination the anatase phase remains stable and peaks intensity decrease. The crystallite size was estimated to be 4.81 nm for $TiO_2$ before calcination and a slight decrease to 4.62 nm was obtained after calcination.

Similar crystallite size (4.19 nm) of anatase $TiO_2$ NPs was previously obtained by other authors [51] using chemical synthesis involving acetylacetone, n-butanol, and 4-dodecylbenzene sulfonic acid in a complex and time-consuming procedure. More eco-friendly method involving the use *S. platensis* living culture managed to obtain crystallite size of 17.3 nm [23]. The present study managed to produce small crystallite size of anatase $TiO_2$ NPs while using facile and environmentally friendly method.

The obtained XRD spectrum of $Co(OH)_2$ NMs (Fig 4K) is in well-agreement with hexagonal $Co(OH)_2$ with COD ID code 1010267, even with secondary metabolites as in the other synthesized materials. The calcination resulted in phase transition to $Co_3O_4$ NMs which matches with cubic $Co_3O_4$ with COD ID code 9005897. The average crystallite sizes of NMs were estimated to be 3.52 and 13.28 nm for $Co(OH)_2$ and $Co_3O_4$, respectively.

Previous studies reported the formation of $Co(OH)_2$ NMs by means of a chemical method utilizing mainly hydrazine or hydrazine hydrate which resulted in crystallite sizes of 2.1 nm and 12.4 nm, respectively [52, 53]. Green synthesized $Co(OH)_2$ NMs were obtained by using *Litchi cinensis* fruit extract with crystallite size of 24.25 nm and 30.28 nm when following the boiling or the microwave route, respectively [54]. While red algae extract was used to obtain $Co_3O_4$ NMs with crystallite size 26.5 nm, only hydrothermal method managed to produce crystallite size of 13.8 nm like the current study [55, 56]. It should be noted that, to best of our knowledge, this work represents the first contribution in the literature where Co NPs were synthesized using a cyanobacterial extract to promote reduction, nucleation and capping.

Morphology of the obtained NMs was studied using SEM and EDX techniques (Fig 4). Further SEM images of the prepared NMs are included in the S1–S3 Figs.

The findings revealed oval shape of Ag NPs before calcination with size range 58–72 nm (Fig 4B) with confirmed presence of silver as well as carbon, oxygen, phosphorus, sulphur and chloride from the *S. platensis* extract (Fig 4D). Even smaller Ag NPs can be seen in the images in the S1 Fig with size of 40 nm. After calcination the agglomerates of Ag NPs having size in the range 72–413 nm can be observed in Fig 4C. The EDX analysis showed the presence of similar elements as before calcination except sulphur (Fig 4E).

Similar spherical shape was observed for $TiO_2$ NPs having size 60–88 nm (Fig 4G). In addition, the sample show high tendency of NPs to form agglomerates of various sizes while the lowest observed size of single NP was measured at 24 nm as shown in the S2 Fig. The EDX analysis confirmed the presence of titanium, carbon, oxygen, nitrogen, phosphorus, and sulphur in their structure (Fig 4I). Calcination slightly increased the size of $TiO_2$ NPs to 63–93 nm (Fig 4H). Moreover, the same agglomeration tendency earlier discussed was observed for calcined samples. In the latter ones, the smallest size of 20 nm was detected (S2 Fig). The same elements were detected with EDX technique as before calcination (Fig 4J).

Interestingly, nanoflake shape was observed for Co(OH)$_2$ NMs with 12–21 nm width and around 500 nm length (Fig 4L). The EDX analysis showed the presence of cobalt, carbon, oxygen, phosphorus, sulphur, and chloride in their structure (Fig 4N).

After phase transition to Co$_3$O$_4$ NMs during calcination, the nanoflake shape is maintained with around 20 nm width (Fig 4M). The EDX analysis confirmed cobalt, carbon, oxygen, and phosphorus in their structure (Fig 4O). To the best of our knowledge, it is the first time nanoflake shape of Co(OH)$_2$ NMs and Co$_3$O$_4$ NMs was obtained using green chemistry method.

The results further corroborate the involvement of *S. platensis* extract in the synthesis of various nanomaterials. Moreover, the products have various shapes and size ranges which show a great potential of secondary metabolites to fabricate different morphologies of nanomaterials.

Thermal properties of the materials were measured in air by both thermal gravimetric analysis (TGA) and differential thermal analysis (DTA) curves (Fig 5). In addition, decomposition temperatures at which the materials lose 5% of their mass, defined as Td, are shown in Table 2.

In case of Ag NPs before (Fig 5A) and after calcination (Fig 5B) three main stages can be observed, i.e. 25–100˚C, 100–600˚C and 600–1000˚C. The slight weight decreases at 952˚C

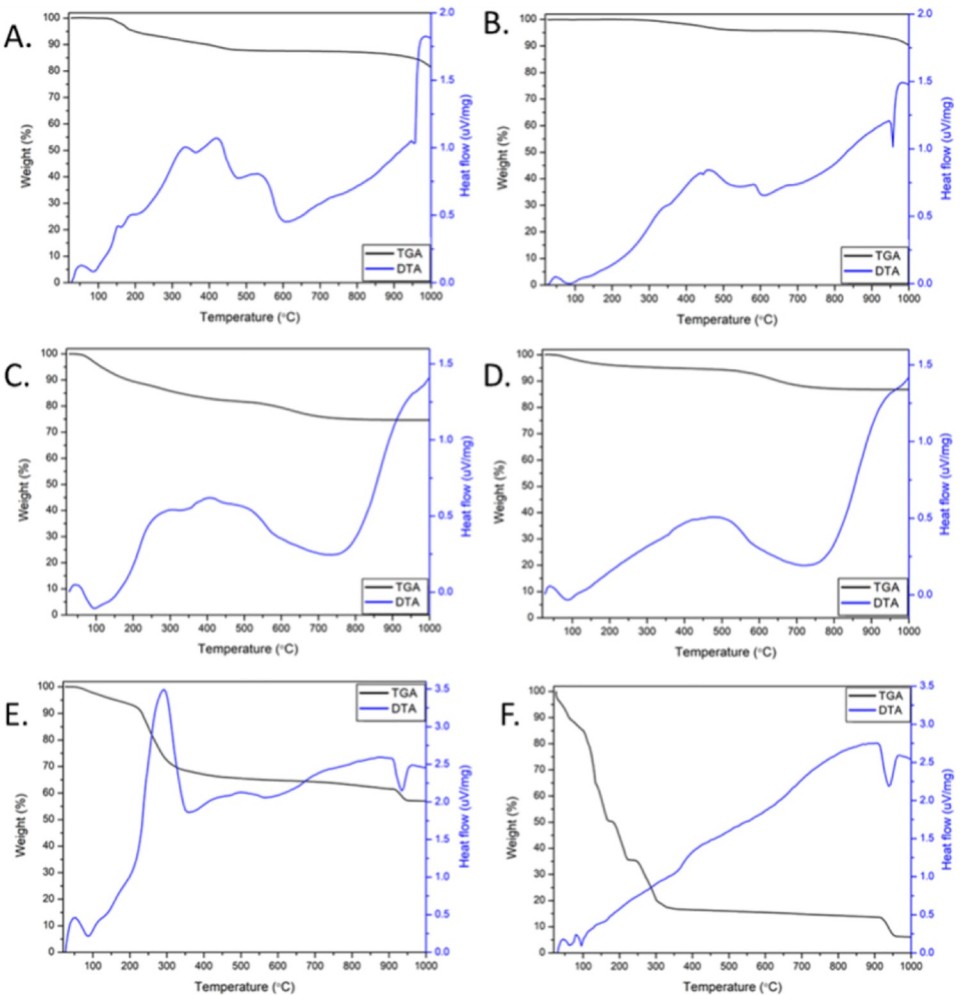

**Fig 5. Thermogravimetric analysis of the prepared nanomaterials.** (A) Ag NPs before calcination, (B) Ag NPs after calcination, (C) TiO2 NPs before calcination, (D) TiO2 NPs after calcination, (E) Co(OH)2 NMs (before calcination), (F) Co3O4 NMs (after calcination).

**Table 2. Decomposition temperatures of the synthesized nanomaterials.**

| | Ag NPs before calcination | Ag NPs after calcination | $TiO_2$ NPs before calcination | $TiO_2$ NPs after calcination | $Co(OH)_2$ NMs before calcination | $Co_3O_4$ NMs after calcination |
|---|---|---|---|---|---|---|
| **Td (°C)** | 197 | 842 | 112 | 359 | 161 | 39 |

Td, decomposition temperatures at which the materials lose 5% of their mass

AgNPs, Silver Nanoparticles; $TiO_2$ NPs, titanium dioxide nanoparticles; $Co(OH)_2$ NMs, cobalt hydroxide nanomaterials; $Co_3O_4$ NMs, cobalt oxide nanomaterials.

when lowering the heat flow corresponds to the melting point of silver. At the first stage, DTA curve indicates exothermal reactions possibly due to the activity of temperature-sensitive compounds at the surface, however with only slight mass difference (weight around 99.9%) in both materials. The progressive temperature increased results in decomposition of secondary metabolites present on Ag NPs surface with heat release and weight observed at 87.58% and 95.76% for Ag NPs before and after calcination, respectively. Above 600°C, loss of molecular oxygen is shown and at the end of the analysis the weight of the samples was 81.41% and for Ag NPs before and after calcination, respectively. In addition, Td values were measured at 197°C for Ag NPs before calcination and increased up to 842°C after calcination. Such difference in Td and the percentage of weight loss can be ascribed to the partial loss or structure changes of secondary metabolites present on the surface due to calcination procedure.

Thermal stability of Ag NPs has been researched and showed final weight at around 70% at 800°C with around 60% weight at 600°C in two separate studies [42, 57]. The research on green synthesized Ag/AgCl nanoparticles from *Oedera genistifolia* plant showed similar 70% weight at 900°C [58]. The improved thermal stability of Ag NPs before and after calcination synthesized from *S. platensis* in the current study further confirms the importance of secondary metabolites on the surface in the achievement of thermally stable nanomaterials.

Similar stages of thermal sensitivity can be observed in $TiO_2$ NPs before (Fig 5C) and after calcination (Fig 5D). Interestingly, until 100°C endothermic loss of water occurs which changed the weight of $TiO_2$ NPs to 96.18% and to 98.33% before and after calcination, respectively. Next, in the 100–750°C interval, the exothermic decomposition of secondary metabolites is observed for $TiO_2$ NPs with weight at 75.31% and 87.54% before and after calcination, respectively. Further temperature increase resulted in loss of molecular oxygen and the final weight at 1000°C was recorded at 74.65% and 86.81% for $TiO_2$ NPs before calcination and after calcination, respectively. Moreover, Td values for $TiO_2$ NPs were observed at 112°C and 359°C before and after calcination, respectively.

The TGA studies on $TiO_2$ NPs synthesized via green routes involving plants, report final weights of about 85% at 950°C and 76.74% at 800°C after calcination at 500°C [59, 60]. The current findings show good thermal stability as well as difference in weight loss between samples before and after calcination similar to Ag NPs 10%. Moreover, a broader DTA peak at 150–750°C in $TiO_2$ NPs reveals a difference in the exothermal reactions taking place likely due to possible distinct bonding of secondary metabolites with metals.

Thermal behaviour was also measured for $Co(OH)_2$ NMs (Fig 5E) and $Co_3O_4$ NMs (Fig 5F). Both materials show mass loss at 25–100°C corresponding to exothermal reactions of temperature-sensitive metabolites, resulting in weights at 97.73% for $Co(OH)_2$ NMs and 85.06% for $Co_3O_4$ NMs. Then, at around 300°C, $Co(OH)_2$ NMs exhibit partial oxidation to $Co_3O_4$ NMs which may be also accompanied by the loss or structural changes of secondary metabolites on the surface after which the weight was recorded at 68.23%. Subsequently, at around 600°C loss of molecular oxygen is shown and the final weight for $Co(OH)_2$ NMs at 1000°C was recorded at 56.98%. Prepared $Co_3O_4$ NMs display exothermal degradation of secondary

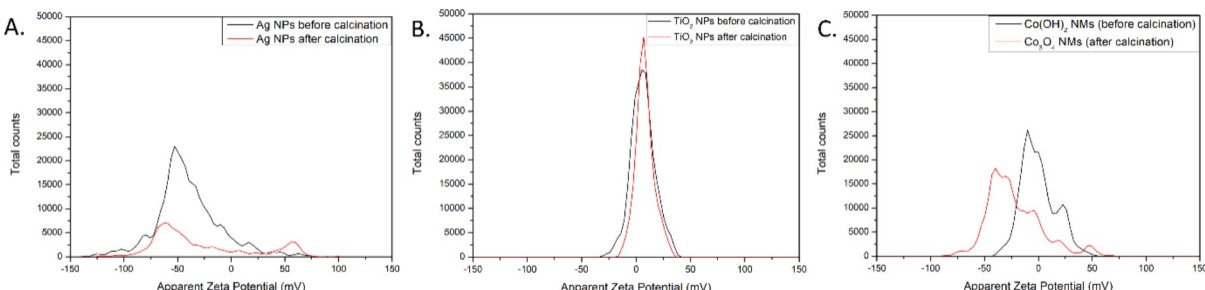

**Fig 6. Zeta potential measurements of the prepared nanomaterials.** (A) Ag NPs before and after calcination, (B) TiO2 NPs before and after calcination, (C) Co(OH)2 NMs and Co3O4 NMs.

metabolites at 100–300°C with a weight at 20.20% which remained relatively stable when further increasing the temperature even though possible structural changes resulted in heat release. The $Co_3O_4$ NMs weight at the end of the analysis was equal to 6.08%. Both $Co(OH)_2$ NMs and $Co_3O_4$ NMs DTA curves show endothermal peak at around 950°C followed by a mass loss which is probably due to reaction with $Al_2O_3$, component of crucible used during the analysis, resulting in appearance of blue colour as it was reported in the literature [61]. In case of $Co(OH)_2$ NMs and $Co_3O_4$ NMs Td values were recorded at 161°C and 39°C, respectively. The decrease in Td values is probably due to change in thermal behaviour of the materials possibly related to the phase transition as it was observed in the XRD analysis.

So far, green synthesized from plants, quasi-spherical $Co_3O_4$ NMs were reported to have around 83% weight at 980°C [62]. The significant weight loss observed during current TGA analysis of $Co(OH)_2$ NMs and $Co_3O_4$ NMs in comparison with other NMs can be attributed to their distinct high content of organic metabolites and unique shape.

Zeta potential measurement was performed to analyse the stability of prepared nanomaterials in water suspension and compare their electrostatic repulsion or attraction behaviour (Fig 6 and Table 3). The findings revealed average zeta potential values reported in Table 3 along with the corresponding standard deviation.

Good stability in water suspension was noted for Ag NPs before calcination which decreased to moderate after calcination procedure. In the literature, zeta potential was measured for Ag NPs before calcination synthesized using water and methanolic *S. platensis* extract revealing the values of -15.90 mV and -43.60 mV, respectively [63, 64]. Most probably the composition of extracted secondary metabolites from S. *platensis* can also play a role in stabilizing NMs in water suspension.

The opposite charge on the surface with poor stability was noted for $TiO_2$ NPs before and after calcination. The rapid ability to coagulate, possibly due to the metabolites on the surface, might be a limitation in efficient action of the nanoparticles. The quick agglomeration was also measured for $Co(OH)_2$ NMs (before calcination) which improved after calcination to $Co_3O_4$ NMs to a relatively poor stability.

**Table 3. Average zeta potential measurements and their standard deviation.**

| | Ag NPs before calcination | Ag NPs after calcination | TiO₂ NPs before calcination | TiO₂ NPs after calcination | Co(OH)₂NMs before calcination | Co₃O₄NMs after calcination |
|---|---|---|---|---|---|---|
| Zeta potential (mV) | -40.2 ± 0.89 | -29.1 ± 2.62 | 6.25 ± 0.32 | 8.05 ± 0.05 | -0.82 ± 1.53 | -22.3 ± 0.51 |

AgNPs, Silver Nanoparticles; $TiO_2$ NPs, titanium dioxide nanoparticles; $Co(OH)_2$ NMs, cobalt hydroxide nanomaterials; $Co_3O_4$ NMs, cobalt oxide nanomaterials.

**Table 4. MIC and MFC determination.**

| | *C. albicans* | | *C. glabrata* | | *C. krusei* | |
|---|---|---|---|---|---|---|
| | MIC (μg/mL) | MFC (μg /mL) | MIC (μg /mL) | MFC (μg /mL) | MIC (μg /mL) | MFC (μg /mL) |
| Ag NPs 1 dark | 50 | 50 | 25 | 50 | 50 | 50 |
| Ag NPs 1 light | 50 | 50 | 50 | 50 | 12 | 25 |
| Ag NPs 2 dark | NE | NE | NE | NE | 50 | 50 |
| Ag NPs 2 light | NE | NE | 100 | 100 | 25 | 25 |
| Co(OH)$_2$ NMs dark | 50 | 100 | 25 | 50 | 25 | 5 |
| Co(OH)$_2$ NMs light | 100 | 100 | 25 | 50 | 12 | 25 |

Ag NPs 1 = before calcination; Ag NPs 2 = after calcination; Co(OH)$_2$ NMs = before calcination; NE = No inhibitory effects on fungal growth.

Statistically significant results have been reported in the results and discussion section.

Six types of NMs: Ag NPs (before and after calcination), TiO$_2$ NPs (before and after calcination), Co(OH)$_2$ NMs, and Co$_3$O$_4$ NMs and relative extracts before synthesis were tested against three *Candida* species to evaluate their inhibitory effect, through broth microdilution technique.

The inhibitory concentrations of nanomaterials having effects on fungal growth ranged from 12 μg/ml to 100 μg/ml. In Table 4, the MIC values of nanomaterials were listed for *C. albicans*, *C. glabrata* and *C. krusei* detected with and without light exposition.

Co$_3$O$_4$ NMs after calcination, TiO$_2$ NPs both before and after calcination and all extracts before synthesis, have no inhibitory activity against *Candida* species under all experimental conditions tested.

The MIC wells presenting lowest growth after 48 hours of incubation were chosen to determine the minimum fungicidal concentration (MFC) against the *Candida* species that was assessed between 25 μg/ml to 100 μg/ml.

Both Ag NPs before calcination and Co(OH)$_2$ NMs (before calcination) showed antifungal activity in all tested conditions and *Candida spp*. Calcination process seems to reduce the antifungal properties of Ag NPs, as the material only managed to inhibit the growth of *C. glabrata* in the presence of light and *C. krusei* in the dark and light. Calcination of Co(OH)$_2$ NMs and subsequent phase transition to Co$_3$O$_4$ NMs significantly changed the properties of the nanomaterial which did not exhibit antifungal properties in the tested concentrations. In addition, TiO$_2$ NPs did not inhibit *Candida* growth before and after calcination in the tested concentrations as well as extract used for the synthesis of NMs. *Candida* species exposed to light, used as a control, showed no alterations in growth.

Considering tested fungal isolates, *C. albicans* was found to be the most resistant to the NMs activity with increased MIC value in the presence of light. Similar increase in the MIC value was detected in case of Ag NPs before calcination treated *C. glabrata*, however, Ag NPs after calcination were found to be antifungal only in the presence of light. The most susceptible fungal pathogen to NMs activity was *C. krusei* which showed decrease in both MIC and MFC values in the presence of light even up to 4 times (MIC of Ag NPs before calcination, p<0.05) (S4–S6 Figs).

The influence of light on antimicrobial activity has been studied before mainly on the bacterial cells [65, 66]. It has been proven that light generates more reactive oxygen species (ROS) which damages the bacterial cells [67]. The recent research on light-activated chemically synthesized NMs found that light can reduce *C. albicans* biofilms (Ag NPs) and inhibit *Fusarium oxysporum* growth (N and F co-doped TiO$_2$ NPs) [68, 69]. However, the light influences not only the performance of NMs, but also the growth of fungi triggering expression of different

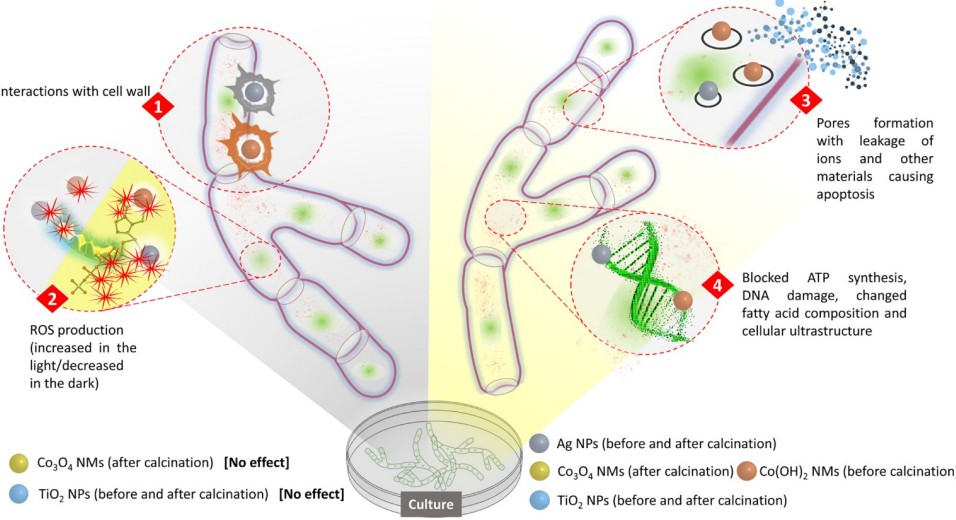

**Fig 7. Hypothesized mechanism of NMs antifungal activity.**

metabolic pathways [70]. In the current study, *C. albicans* exposed to Co(OH)$_2$ NMs (before calcination) and *C. glabrata* exposed to Ag NPs before calcination required increased concentrations of NMs to inhibit fungal growth. The results suggest potential activation of fungal defence mechanisms in the presence of light which requires further studies.

The proposed antifungal mechanism in depicted in Fig 7. The first interactions occur possibly between NMs and fungal cell wall which can cause disintegration and damages also to the cell membrane [48]. The process activates ROS production which is usually enhanced in the presence of light [71]. As a result, the fungal cell exhibits oxidative stress causing formation of pores, leakage of cell content and apoptosis [72]. At the same time, ATP synthesis is blocked, DNA is damaged, fatty acid composition is changed and overall ultrastructure of the cell is disturbed [73]. The exact molecular mechanism leading to above mentioned effects is a result of interactions between cell components and specific structure (size, shape) as well as composition (identified phases) of the synthesized NMs and may differ accordingly [74]. Additionally, Ag NPs can release Ag$^+$ ions which have high affinity and may bind to the thiol group (-SH) of the proteins affecting their original structure and activity (denaturation) [68]. The antifungal properties of Co(OH)$_2$ NMs were not studied before, however, it is probable that due to the presence of hydroxy group (-OH) in their structure they may affect hydrogen bonds formation in the proteins causing structural changes and inhibiting proper protein functions in the cell. The exact mechanism of the antifungal activities of various NMs requires further research.

## Conclusion

The study showed the potential of secondary metabolites from *S. platensis* to participate in the synthesis of various NMs. The involvement of amines among other compounds was revealed by GCMS and FTIR analyses. The phases identification in XRD analysis confirmed the presence of Ag and AgCl phases in Ag NPs, TiO$_2$ anatase in TiO$_2$ NPs, Co(OH)$_2$ in Co(OH)$_2$ NMs and Co$_3$O$_4$ in Co$_3$O$_4$ NMs. The SEM findings showed oval and spherical shapes for Ag NPs and TiO$_2$ NPs, respectively with nanoflake shape of both Co(OH)$_2$ NMs and Co$_3$O$_4$ NMs. EDX further confirmed the presence of organic content on the surface of the prepared NMs. In addition, Ag NPs showed the best thermal stability of all the synthesized NMs which

increased after calcination. It has been found that Ag NPs exhibited the best stability in water suspension, even though it decreased after calcination.

We have also evaluated, for the first time, the capability of six NMs to act against several *Candida* species and our data have supported our hypothesis: Ag NPs (before and after calcination) and $Co(OH)_2$ NMs have shown antifungal activity for all *Candida* tested. Interestingly, these nanostructures exhibited high antimicrobial activity on *C. glabrata* that was amphotericin resistant. It is apparent that the action mechanism of the NPs also depends on light conditions while the underlying molecular processes involved require further investigation.

Our results led us to speculate that both the composition of nanomaterials and the different experimental conditions (presence/absence of light) may have different mechanisms of cytolysis, inducing candidacidal effect through mechanisms already described, such as insertion of functional pores into target membranes, induction of apoptosis, alteration of macromolecules structure, or inhibition of ATP synthesis.

These findings, together with the increase of microbial resistance to antifungal drugs, encourage to carry out further studies to also determine the antibacterial activity against multi-drug resistance pathogens as well as the biocompatibility, cytotoxicity and safety of these nanostructures, with a view to their possible use in the medical field.

## Supporting information

**S1 Fig.** SEM images of Ag NPs (A) before calcination, (B) after calcination.
(TIFF)

**S2 Fig.** SEM images of TiO2 NPs (A) before calcination, (B) after calcination.
(TIFF)

**S3 Fig.** SEM images (A) $Co(OH)_2$ (before calcination), (B) $Co_3O_4$ (after calcination).
(TIFF)

**S4 Fig. Antifungal activities of Ag NPs before and after calcination, $Co(OH)_2$ NMs (before calcination) and $Co_3O_4$ NMs (after calcination) on *C. albicans* in the presence and absence of light.**
(TIFF)

**S5 Fig. Antifungal activities of Ag NPs before and after calcination, $Co(OH)_2$ NMs (before calcination) and $Co_3O_4$ NMs (after calcination) on *C. glabrata* in the presence and absence of light.**
(TIFF)

**S6 Fig. Antifungal activities of Ag NPs before and after calcination and $Co(OH)_2$ NMs (before calcination) on *C. krusei* in the presence and absence of light.**
(TIFF)

**S1 Table. Composition of the samples before and after synthesis of Ag NPs as well as in the mimicking reaction.**
(DOCX)

**S2 Table. Composition of the samples before and after synthesis of $TiO_2$ NPs as well as in the mimicking reaction.**
(DOCX)

**S3 Table. Composition of the samples before and after synthesis of $Co(OH)_2$ as well as in the mimicking reaction.**
(DOCX)

## Acknowledgments

A.S. performed her activity in the framework of the International PhD in Innovation Sciences and Technologies at the University of Cagliari, Italy. The contribution of the latter institution through Prof. Pierluigi Caboni for gas chromatographic and FTIR analyses, Dr Francesco Loy for SEM and EDX characterization as well as Prof. Annamaria Fadda and Dr Carla Caddeo for Zeta potential evaluation is gratefully acknowledged.

## Author Contributions

**Conceptualization:** Agnieszka Sidorowicz, Antonella Pantaleo, Alessandro Concas, Paola Rappelli, Pier Luigi Fiori, Giacomo Cao.

**Data curation:** Agnieszka Sidorowicz, Valentina Margarita, Alessia Manca, Giacomo Cao.

**Formal analysis:** Antonella Pantaleo, Alessandro Concas.

**Investigation:** Agnieszka Sidorowicz, Valentina Margarita, Giacomo Fais, Alessia Manca.

**Methodology:** Agnieszka Sidorowicz, Valentina Margarita.

**Resources:** Giacomo Cao.

**Supervision:** Antonella Pantaleo, Pier Luigi Fiori, Giacomo Cao.

**Validation:** Alessandro Concas, Paola Rappelli, Giacomo Cao.

**Writing – original draft:** Agnieszka Sidorowicz, Valentina Margarita.

**Writing – review & editing:** Pier Luigi Fiori, Giacomo Cao.

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
