## [Decision Letter · Decision Letter 0]

26 Jul 2022

PONE-D-22-13061Characterization of nanomaterials synthesized from Spirulina platensis extract and their potential antifungal activityPLOS ONE

Dear Dr. Fiori,

Thank you for submitting your manuscript to PLOS ONE. After careful consideration, we feel that it has merit but does not fully meet PLOS ONE’s publication criteria as it currently stands. Therefore, we invite you to submit a revised version of the manuscript that addresses the points raised during the review process.

We look forward to receiving your revised manuscript.

Kind regards,

Raghvendra Bohara

Academic Editor

PLOS ONE

Journal Requirements:

Reviewers' comments:

Reviewer's Responses to Questions

**Comments to the Author**

1. Is the manuscript technically sound, and do the data support the conclusions?

Reviewer #1: Yes

Reviewer #2: Yes

2. Has the statistical analysis been performed appropriately and rigorously? 

Reviewer #1: Yes

Reviewer #2: No

3. Have the authors made all data underlying the findings in their manuscript fully available?

Reviewer #1: Yes

Reviewer #2: No

4. Is the manuscript presented in an intelligible fashion and written in standard English?

Reviewer #1: No

Reviewer #2: Yes

5. Review Comments to the Author

Reviewer #1: The work is good and fits in the scope of Plos One. Please address following comments.

L37: This sentence is not clear.

L55: Narrow spectrum of activity

L61: altered physicochemical properties? Superior than bulk materials? Name some biologically important properties here.

L62 and the whole manuscript: In title, authors say nanomaterials, here in L62 it is nanoparticles, and at various places in the manuscript, it is nanostructure. PLease use a unifrom term and that can be nanoparticles (NPs) only since what you have synthesized are NPs only.

L66: This sentence needs a reference. you can see the following paper(s): Advanced Powder Technology 29 (7), 1601-1616; Process Biochemistry 91, 387-397

references in the bibliography are not in a proper style i.e. remove 'Available from' and '[Internet]' from references. In some references, full bibliographic information is not there. Also, please italicize the scientific names.

The introduction can be made compact. For example, some studies are discussed in detail such as Ameen et al. (19) occupies L78-84 and the ref. (20) is discussed from L84-88. Mentioning of these studies can be shortened. Similar should be done with the whole introduction.

L89: You have already abbreviated NPs somewhere before. Please use a single abbreviation throughout.

The ref. 23 is a review paper that is okay to cite. Recently Fatima et al. (2021) have tested TiO2 NPs on oral biofilm forming bacteria and reported significant inhibition. Authors can see this paper and cite: Pharmaceutics 13 (10), 1564

L98: So did you perform any study to check the effects of S. platensis metabolites on TiO2 NPs formation?

L109: This perhaps be not true because in a study, Anwar et al. reports Cobalt nanoparticles as novel nanotherapeutics against Acanthamoeba castellanii (PMID: 31159839)

L116: Only three types, calcination does not change the type but may change physicochemical and/or biological properties.

L119: Write Candida at first instance and then use 'C."

L135: Wavelength for Abs or OD is 755 or 750?

L146: Instead of an asterisk, use dot only.

L147: What was the need to modify Zarrouk medium?

L157: Nanostructures or NPs?

for silver and cobalt salts up to 8, for titanium up to 4 but what is 3-4? Moreover, you have used 1.25 M NaOH to maintain the pH but it could have diluted the molarity of metal salts Ag, Co, and Ti, how did you then maintain the 0.1 M molarity?

L164: Reaction, with stirring or without stirring?

L237: Use abbreviation for Candida.

L238: Please give a reference for previous paper where isolation and characterization of Candida strains is reported.

L242-345: Here you can simply say the OD was measured at 600 nm.

This is the problem I found at several places where unnecessarily the text is more.

L246: 48-well is simply a multi-well plate not microtiter plate since its capacity is more than 1 mL.

L249: Put an space between mg/ml and value.

L257: Who defined the MIC, any original reference?

L265: Abbreviate Spirulina here and other places.

L482: What is the meaning of this difference in zeta-potential? It shows that the overall charge on NPs surface is negative. Also seta values shows how stable the NPs is, please add text for stability.

L586-597 and L601-603: This is not conclusion, these are results only. Authors should rewrite the conclusion

Figure 7: Authors should hypothesize a mechanism based on the results reported in this study only. Was interactions of NPs with cell wall, and pore formation, apoptosis, ATP synthesis, changed in cellular structure and fatty acid composition were checked by the authors?

Reviewer #2: The manuscript introduces the details on the synthesis and full characterization of a number of nanomaterial structures using a green chemistry approach, with evaluating the activity of candidate particles against three Candida species. Some suggestions are listed below for enhancing the presentation of the work:

Abstract

-It is better to involve one or two sentences on the choice of the particular tested nanoparticles.

-The last sentence: should be splitted into two or more sentences with giving the exact values of MIC and the meaning of this abbreviation.

Introduction

-Very long section, specially the Ag NPs part, and should be summarized with transferring of a part to the discussion section.

Methodology

-Paragraph starting with (It should be noted that the trace …..) should be moved to the first paragraph under this section after the sentence beginning with (The medium was prepared according….).

-Synthesis of nanostructures: How much metal salts were added to the extract to synthesize each type of particles? Were they added as powder or in solution?

-Materials characterization: The sentence beginning with (Then, a 1.25 M HCl solution was ….) is not clear. Was the initial pH in case of titanium NPs zero?

Results

-Generally, the units are inconsistent between tables and the main text. No statistical analysis was performed, particularly for the experiments assessing the inhibitory concentrations of nanomaterials.

-Fungal study & Table 4: There are no details about any control groups studied. Is there any effect of light only on the tested fungi?

-The supplementary figures & tables are not present in the uploaded version.

6. PLOS authors have the option to publish the peer review history of their article (what does this mean?). If published, this will include your full peer review and any attached files.

Reviewer #1: **Yes: **Bilal Ahmed

Reviewer #2: **Yes: **Amir Alsharabasy

---

## [Author Response · Author response to Decision Letter 0]

12 Aug 2022

First, we would like to thank the Reviewers for their thorough work in revising the manuscript and for their valuable advice to improve and clarify our work. In this list we clarify all the critical points highlighted by the reviewers.

Reviewer #1: The work is good and fits in the scope of Plos One. Please address following comments.

We thank you very much for your appreciation of our work!

1. L37: This sentence is not clear.

Thank you for the comment. Explanation of “Minimum Inhibitory Concentration (MIC)”has been added for better understanding of the sentence.

2. L55: Narrow spectrum of activity

Thank you for the notice. The sentence was corrected.

3. L61: altered physicochemical properties? Superior than bulk materials? Name some biologically important properties here.

Thank you for the inquiry. Prepared nanostructures often exhibit different properties than their bulk state equivalents. Some of the different physicochemical properties include: increased surface to volume ratio, high reactivity, altered magnetic and optical properties. The explanation was added in the text.

4. L62 and the whole manuscript: In title, authors say nanomaterials, here in L62 it is nanoparticles, and at various places in the manuscript, it is nanostructure. PLease use a unifrom term and that can be nanoparticles (NPs) only since what you have synthesized are NPs only.

Thank you for the comment. We followed the International Organization for Standardization (IOS) standards (ISO/TS 12901-1:2012) specifying the difference between nanoparticle and nanostructure. In the document, nanoparticle is defined as a 3D object with similar axes (spherical shape). Since not all the synthesized nanostructures follow that shape, it is more precise to differentiate between nanoparticles (such as Ag NPs) and nanostructures/nanomaterials (such as Co(OH)2 NMs which have nanoflake morphology).

5. L66: This sentence needs a reference. you can see the following paper(s): Advanced Powder Technology 29 (7), 1601-1616; Process Biochemistry 91, 387-397

Thank you for the remark. The above references have been added to the manuscript

6. references in the bibliography are not in a proper style i.e. remove 'Available from' and '[Internet]' from references. In some references, full bibliographic information is not there. Also, please italicize the scientific names.

Thank you for the notice. The references were corrected accordingly

7. The introduction can be made compact. For example, some studies are discussed in detail such as Ameen et al. (19) occupies L78-84 and the ref. (20) is discussed from L84-88. Mentioning of these studies can be shortened. Similar should be done with the whole introduction.

Thank you for the suggestions. The introduction was shortened as suggested. 

8. L89: You have already abbreviated NPs somewhere before. Please use a single abbreviation throughout.

Thank you for the notice. The line was corrected.

9. The ref. 23 is a review paper that is okay to cite. Recently Fatima et al. (2021) have tested TiO2 NPs on oral biofilm forming bacteria and reported significant inhibition. Authors can see this paper and cite: Pharmaceutics 13 (10), 1564

Thank you for the comment. The above reference was added to the revised manuscript.

10. L98: So did you perform any study to check the effects of S. platensis metabolites on TiO2 NPs formation?

Thank you for the comment. The description of the metabolites and their participation in the TiO2 NPs formation has been added, as suggested. 

11. L109: This perhaps be not true because in a study, Anwar et al. reports Cobalt nanoparticles as novel nanotherapeutics against Acanthamoeba castellanii (PMID: 31159839)

Thank you for the suggestion. The cited study used hydrothermal method (chemical synthesis) to synthesize cobalt nanostructures which belongs to the different group of synthesis methods. So far, the action of green-synthesized cobalt hydroxide against fungal pathogens is not well-studied. The changes were made in the text and the paper by Anwar et al. was added and discussed. 

12. L116: Only three types, calcination does not change the type but may change physicochemical and/or biological properties.

Thank you for the remarks. The calcination indeed may change physicochemical and/or biological properties. However, the reported XRD analysis show difference in the obtained pattern before and after calcination which signifies that also difference in the type of material occurred.

13. L119: Write Candida at first instance and then use 'C."

Thank you for the notice. The sentence was corrected.

14. L135: Wavelength for Abs or OD is 755 or 750?

Thank you for the comment. The wavelength was corrected to 750 nm

15. L146: Instead of an asterisk, use dot only.

Thank you for the suggestion. The text was modified accordingly.

16. L147: What was the need to modify Zarrouk medium?

Thank you for the question. The medium was modified to enhance biomass production of Spirulina platensis. The explanation was added in the text.

17. L157: Nanostructures or NPs?

Thank you for the notice. As mentioned above, the sentence refers to nanostructures, as not all the prepared materials had spherical shape (according to IOS standards ISO/TS 12901-1:2012).

18. for silver and cobalt salts up to 8, for titanium up to 4 but what is 3-4? Moreover, you have used 1.25 M NaOH to maintain the pH but it could have diluted the molarity of metal salts Ag, Co, and Ti, how did you then maintain the 0.1 M molarity?

Thank you for the comment. As explained in the description of the mimicking reaction, after adding the salt, the pH dropped to 5 for silver and cobalt and 0 for titanium. Subsequently, it was raised for silver and cobalt salts up to 8, for titanium up to 4. The changes in the manuscript were made to clarify the reaction. Moreover, the molarity of the metal salts was not adjusted after adding them to the extract.

19. L164: Reaction, with stirring or without stirring?

Thank you for the remark. The reaction was conducted with stirring (250 RPM) and heating (85°C), as described at line 159. The changes were introduced in the manuscript.

20. L237: Use abbreviation for Candida.

“Candida” was abbreviated, as suggested

21. L238: Please give a reference for previous paper where isolation and characterization of Candida strains is reported.

The microorganisms were isolated and identified in our reference laboratory. The molecular identification techniques of Candida species used have been included and discussed in the text (MALDI-TOF analysis of pure culture isolated in selective media)

22. L242-345: Here you can simply say the OD was measured at 600 nm.

This is the problem I found at several places where unnecessarily the text is more.

Thank you for the comment. The sentence has been corrected and the text has been revised in many of its parts

23. L246: 48-well is simply a multi-well plate not microtiter plate since its capacity is more than 1 mL.

Thank you for the comment. The corrections were made.

24. L249: Put an space between mg/ml and value.

Space was inserted, as suggested

25. L257: Who defined the MIC, any original reference?

A specific reference was included in the text (reference 32)

26. L265: Abbreviate Spirulina here and other places.

Thank you for the comment. The corrections were made.

27. L482: What is the meaning of this difference in zeta-potential? It shows that the overall charge on NPs surface is negative. Also seta values shows how stable the NPs is, please add text for stability.

Thank you for the remark. The description of differences and also stability, is discussed in the text. Moreover, the possible explanation of the phenomena is given in revised version .

28. L586-597 and L601-603: This is not conclusion, these are results only. Authors should rewrite the conclusion

Thank you for the comment. The mentioned part shortly summarizes main findings of the study for better understanding of the work, which in the latter part of the text leads to the future perspective based on the obtained results. The Conclusions have been rewritten and shortened as suggested, also including hypotheses on the mechanism of growth inhibition, and killing of Candida species

29. Figure 7: Authors should hypothesize a mechanism based on the results reported in this study only. Was interactions of NPs with cell wall, and pore formation, apoptosis, ATP synthesis, changed in cellular structure and fatty acid composition were checked by the authors?

Thank you for the comment. The exact mechanism of the action is very interesting, as it differs not only depending on the type of the nanomaterials used and species of pathogens, but also on the applied conditions (light/no light). So far, the above-mentioned changes were observed in the literature and similar investigations require dedicated study which we will consider in the future. Some hypotheses regarding the mechanism of action have been included in the text

Reviewer #2: The manuscript introduces the details on the synthesis and full characterization of a number of nanomaterial structures using a green chemistry approach, with evaluating the activity of candidate particles against three Candida species. Some suggestions are listed below for enhancing the presentation of the work:

Thank you very much for the suggestions that really made it possible to improve our work

Abstract

1. -It is better to involve one or two sentences on the choice of the particular tested nanoparticles.

Thank you for the comment. The choice is explained in the introduction, proper description of the choice would extend the word limit for abstract.

2. -The last sentence: should be splitted into two or more sentences with giving the exact values of MIC and the meaning of this abbreviation.

Thank you for the remark. The sentence was corrected as suggested.

Introduction

3. -Very long section, specially the Ag NPs part, and should be summarized with transferring of a part to the discussion section.

Thank you for the suggestion. The introduction (specially the Ag NPs sentences) was shortened as suggested.

Methodology

4. -Paragraph starting with (It should be noted that the trace …..) should be moved to the first paragraph under this section after the sentence beginning with (The medium was prepared according….).

Thank you for the remark. The paragraph was moved accordingly.

5. -Synthesis of nanostructures: How much metal salts were added to the extract to synthesize each type of particles? Were they added as powder or in solution?

Thank you for the inquiries. The metal salts were added in powder and their exact weight was added to the text.

6. -Materials characterization: The sentence beginning with (Then, a 1.25 M HCl solution was ….) is not clear. Was the initial pH in case of titanium NPs zero?

Thank you for the comment. After the salt addition, the pH drops (3 points for silver and cobalt, 4 points for titanium) due to the acidic nature of the used salts. In case of titanium it dropped to zero as stated in the text.

Results

7. -Generally, the units are inconsistent between tables and the main text. No statistical analysis was performed, particularly for the experiments assessing the inhibitory concentrations of nanomaterials.

Thank you for your comment: the error was due to a missed check in the final submitted text. In the new version the inconsistencies between the text and the tables have been corrected. Statistical analysis of the results has been included in the revised version

8. -Fungal study & Table 4: There are no details about any control groups studied. Is there any effect of light only on the tested fungi?

The results of the control groups were reported in the text (in particular, the absence of effect on fungal growth by light alone)

9. -The supplementary figures & tables are not present in the uploaded version.

We apologize for the mistake: supplementary tables and figures have been now included in the revised version

---

## [Editor Report · Decision Letter 1]

6 Sep 2022

Characterization of nanomaterials synthesized from Spirulina platensis extract and their potential antifungal activity

PONE-D-22-13061R1

Dear Dr. Fiori,

We’re pleased to inform you that your manuscript has been judged scientifically suitable for publication and will be formally accepted for publication once it meets all outstanding technical requirements.

Kind regards,

Raghvendra Bohara

Academic Editor

PLOS ONE

Additional Editor Comments (optional):

I am satisfied with the revised version.
---

## [Editor Report · Acceptance letter]

7 Sep 2022

PONE-D-22-13061R1 

Characterization of nanomaterials synthesized from *Spirulina platensis* extract and their potential antifungal activity 

Dear Dr. Fiori:

I'm pleased to inform you that your manuscript has been deemed suitable for publication in PLOS ONE. Congratulations! Your manuscript is now with our production department. 

Kind regards, 

on behalf of

Dr. Raghvendra Bohara 

Academic Editor

PLOS ONE